

# Bridges in social networks: current status and challenges

Jeongseon Kim[1,2], Soohwan Jeong[1], Jungeun Kim[3] and Sungsu Lim[1]

[1] Department of Computer Science and Engineering, Chungnam National University, Daejeon, Republic of Korea
[2] Fiscal and Economic Policy Intelligence Research Center, Electronics and Telecommunications Research Institute, Daejeon, Republic of Korea
[3] Department of Computer Engineering, Inha University, Incheon, Republic of Korea

## ABSTRACT

In social network analysis, bridges play a critical role in maintaining connectivity and facilitating the dissemination of information between communities. Despite increasing interest in bridge structures, a systematic classification of their roles across various network types remains unexplored. This study introduces a categorization of bridges into structural and functional types. Structural bridges maintain connectivity by preventing network fragmentation, whereas functional bridges facilitate the flow of information between communities. We conducted a comprehensive literature review and classified existing studies within this framework. The findings clarify the distinct roles of bridges and provide valuable insight for devising effective strategies for network design and analysis.

## INTRODUCTION

Social networks have become a crucial area of study due to their ability to model complex relationships and interactions between individuals, organizations, and systems (*Watts, 2004a*, *2004b*; *Easley & Kleinberg, 2010*). These networks encompass diverse relationships, ranging from collaborations to rivalries, providing versatile frameworks to understand human behavior and dynamics (*Easley & Kleinberg, 2010*; *Roth & Ockenfels, 2002*). Analyzing social networks provides valuable insights into the flow of information, influence, and resources within and across communities, with significant implications for fields such as sociology, economics, public health, and computer science. By examining the nodes and edges of social networks, researchers can uncover patterns, identify influential nodes, and evaluate structural robustness (*Bonacich, 1987*; *Freeman, 1978*; *Albert & Barabási, 2002*). Understanding the structural and functional components of these networks is essential to enhance their efficiency, resilience, and overall functionality.

Social networks are widely analyzed across various domains to uncover meaningful insights and address complex challenges. In marketing, key influencers are identified to optimize the dissemination of promotional content (*Watts & Dodds, 2007*). In healthcare, interaction patterns within populations are examined to track the spread of infectious diseases (*Christakis & Fowler, 2007*). Social media platforms leverage network analysis to

Corresponding author
Sungsu Lim, sungsu@cnu.ac.kr

recommend personalized content and detect community structures (*Tang, Aggarwal & Liu, 2016*). Similarly, critical links between genes or proteins are revealed in biological networks, providing valuable insights into disease mechanisms and potential therapeutic targets (*Barabasi & Oltvai, 2004*). In transportation networks, structural bridges enhance connectivity and prevent cascading disruptions in urban transit systems (*Xiao, Huang & Tang, 2023*; *Qiang, Wancheng & Xinzhi, 2021*). Likewise, in energy networks, bridge-block structures are optimized to locate disruptions and improve grid stability (*Lan & Zocca, 2021*; *Ódor et al., 2024*). These diverse applications highlight the versatility and importance of social network analysis in multiple domains. While the concept of bridges applies across diverse network types, this review primarily focuses on social networks. Applications to biological, transportation, and energy networks are discussed separately in 'Applications In Other Domains'.

Bridges are fundamental concepts in social network analysis (*Granovetter, 1973*; *Papadopoulos et al., 2009*; *Easley & Kleinberg, 2010*; *Zhang & Li, 2017*). A bridge is typically defined as an edge whose removal increases the number of disconnected components in the network, thus linking parts that would otherwise remain separate. More generally, a bridge can refer to a connection, either an edge or a node, that links distinct subcomponents of a network. These connections play a crucial role in maintaining network connectivity (*Granovetter, 1973*). Understanding the roles of bridges provides valuable information on the interactions between different parts of the network (*Granovetter, 1973*; *Burt, 2005*; *Papadopoulos et al., 2009*). Analyzing such connections also helps identify vulnerabilities that could result in network fragmentation (*Albert & Barabási, 2002*; *Borgatti & Everett, 2006*). Insights obtained from bridge analyses inform strategies to improve network resilience and improve information flow efficiency (*Burt, 2005*; *Newman, 2003*).

In addition to bridges, other fundamental concepts, such as hubs and structural holes, provide complementary perspectives for understanding network structures. Hubs (*Barabási & Albert, 1999*; *Freeman, 1978*; *Kim, Kwon & Lee, 2017*; *Borgatti & Everett, 2006*), structural holes (*Burt, 1992*, *2005*; *Burt & Soda, 2021*; *Lin et al., 2021*), and bridges constitute core concepts in network structure analysis. Hubs act as centers of connectivity, structural holes create opportunities for intermediaries, and bridges serve as critical connections that connect distinct communities, enabling the flow of information across otherwise isolated regions. These concepts provide insights into distinct but interconnected roles that shape the dynamics of social networks. This article systematically explores bridges, highlighting their unique roles and contributions to network connectivity and information flow.

This survey introduces a categorization of bridges, distinguishing them as structural bridges and functional bridges. Structural bridges represent critical connections in a network whose removal leads to structural fragmentation, disconnecting previously connected components. Functional bridges, in contrast, refer to links or nodes that facilitate the flow of information and resources between communities, even when the overall network remains structurally intact. The categorization provides a deeper understanding of the distinct roles that structural and functional bridges play in social

| Table 1 Comparison of structural and functional bridges. | | |
|---|---|---|
| **Feature** | **Structural bridge** | **Functional bridge** |
| Purpose | Maintains network connectivity | Supports efficient information flow |
| Effect of removal | Causes network fragmentation | Reduces efficiency of information diffusion |
| Focus | Structural robustness | Information propagation |
| Typical identification methods | Betweenness centrality, articulation points | Flow-based metrics, influence spread analysis |

networks. While structural bridges ensure network connectivity, functional bridges improve the efficiency of information flow and resource distribution. Table 1 summarizes the purpose, effects of removal, focus, and common identification methods for structural and functional bridges, based on a comprehensive analysis of existing literature.

While prior surveys have extensively examined community detection (*Fortunato & Hric, 2016*), structural hole theory (*Burt, 2005*), and node centrality measures (*Freeman, 1978*), few have provided a systematic synthesis focused specifically on bridges as distinct structural and functional entities. Existing reviews often treat bridges as peripheral aspects of broader topics rather than as primary constructs deserving dedicated analysis. In contrast, this survey contributes a novel perspective by consolidating diverse lines of research into a unified framework that distinguishes structural and functional bridges, integrates insights from both signed and unsigned networks, and contextualizes recent advances in graph embedding and dynamic network analysis. This approach clarifies conceptual boundaries, highlights underexplored research opportunities, and underscores the practical relevance of bridges across domains such as infrastructure resilience, information diffusion, and online social platforms.

The remainder of the article is organized as follows. 'Survey Methodology' details the literature search strategy, selection criteria, and measures to ensure unbiased study inclusion on bridges in social networks. 'Preliminaries' covers basic concepts in social network analysis, including the distinction between signed and unsigned networks, and the role of bridges. 'Bridge Properties' discusses the details of structural and functional bridges. 'Research Areas reviews' three key research areas where bridges are important, along with key contributions. 'Applications In Other Domains' outlines the practical applications of bridges. 'Opportunities and Challenges' addresses challenges, opportunities, and future research directions. 'Conclusion' summarizes the findings and concludes with remarks on the importance of bridges in network analysis.

## SURVEY METHODOLOGY

This study follows a systematic approach to ensure comprehensive and unbiased coverage of the literature on bridges in social networks. Our methodology involved structured search strategies, inclusion criteria, and a review process that aligns with best practices for literature surveys.

## Literature search strategy

We performed a systematic search across multiple academic databases, including Google Scholar, Web of Science, and Scopus to identify relevant studies. We used a combination of targeted search queries based on relevant keywords such as "bridge detection in social networks," "network connectivity bridges," "functional and structural bridges," and "bridge nodes in signed networks." Additionally, we reviewed references from key survey articles to ensure the inclusion of influential studies that are not always identified through keyword searches.

We applied a structured literature review methodology to ensure the reliability of our search process. The study selection process includes the removal of duplicate articles, an initial title and abstract screening, and a full-text review to determine relevance to the research scope. Studies that did not explicitly define or apply bridges in social networks were excluded.

## Inclusion and exclusion criteria

To ensure a high standard of relevance and quality, the following **inclusion criteria** are defined:

- Studies that explicitly examine bridge detection and its structural or functional aspects, or their applications in network analysis.
- Peer-reviewed journal and conference articles, as well as preprints available in recognized repositories, published from 2,000 onward.
- Although the main inclusion window began in 2,000, key foundational studies predating this period were included because they established essential concepts relevant to subsequent research.
- Articles that present empirical analysis, formal definitions, or algorithmic approaches to bridge detection.
- Articles that address bridges in both signed and unsigned networks.

We applied the following **exclusion criteria**:

- Studies that do not provide an explicit definition or application of bridges in social networks.
- Articles that focus solely on community detection without discussing bridges.
- Duplicates retrieved from multiple databases.

The final selection process involved multiple screening steps, including title and abstract review, full-text examination, and duplicate removal, to ensure the inclusion of the most relevant studies.

## Ensuring comprehensive and unbiased coverage

We adopted several strategies to maintain a balanced and unbiased literature review. First, we included studies from multiple domains, such as computational social science, network theory, and applied machine learning, to prevent disciplinary bias. Additionally, we

ensured a mix of foundational works (*e.g.*, (*Granovetter, 1973*; *Burt, 2005*)) and recent developments in bridge detection. We implemented clearly defined selection rules and cross-validated our study inclusion against existing review articles on social network bridges to enhance objectivity. This approach minimizes the overrepresentation of specific perspectives while maintaining a comprehensive review.

This survey presents a comprehensive and balanced analysis of bridge structures in social networks, incorporating key theoretical and empirical contributions. Furthermore, the categorization framework developed in this study is expected to serve as a valuable reference for future research in bridge detection algorithms and network analysis. This structured review supports the continued advancement of methodologies for identifying and characterizing bridges across different network types.

A total of 24 core studies were included in this review (20 on unsigned networks and four on signed networks). Additionally, 12 application-oriented articles were selected to illustrate practical implementations of bridge concepts in diverse domains, including medicine, health and behavioral sciences, biology, transportation, and energy systems.

# PRELIMINARIES

In this section, we introduce key concepts in network analysis to establish a foundation for understanding bridges in social networks. Positive and negative relationships influence how bridges are interpreted. Distinguishing between signed and unsigned networks is crucial for accurate analysis. Accordingly, this survey categorizes networks into unsigned and signed types based on this distinction.

## Basic concepts in network analysis

Network analysis examines the structural and functional properties of networks, which are typically modeled as graphs consisting of vertices and edges (*Newman, 2010*; *Easley & Kleinberg, 2010*). These graphs vary in complexity, featuring directed or undirected edges and weighted or unweighted links, depending on the nature of the relationships (*Watts, 2004a*). Directed edges represent asymmetric relationships, such as one-way communication or influence, while undirected edges indicate mutual interactions. Weighted edges reflect the strength or importance of connections, whereas unweighted edges assume uniform relationships. Understanding these distinctions is essential for accurately modeling real-world networks and analyzing the roles of bridges across various graph types.

Key concepts in network analysis include: (i) degree, which represents the number of edges connected to a node, with high-degree nodes often functioning as hubs (*Newman, 2010*; *Barabási & Albert, 1999*); (ii) paths, sequences of edges that connect nodes, crucial for understanding network efficiency (*Newman, 2010*); (iii) connected components, groups of nodes that are mutually reachable from each other (*Newman, 2010*; *Erdös & Rényi, 1959*); and (iv) centrality measures, such as degree centrality and betweenness centrality, which quantify the importance of nodes within the network (*Freeman, 1978*; *Borgatti & Everett, 2006*; *Bonacich, 1987*).

## Unsigned networks

Unsigned networks are characterized by edges that lack a sign, representing relationships without categorizing them as positive or negative. These networks are used when the nature of the relationship is neutral, unknown, or irrelevant to the analysis. Formally, an unsigned network is defined as $G = (V, E)$, where $V$ denotes the set of vertices, and $E$ represents the set of edges. The focus in unsigned networks is primarily on the structural properties rather than the polarity of relationships. Although unsigned networks have primarily been used to study structural properties, they have also served as frameworks for analyzing polarization phenomena (*Morales et al., 2015*; *Garimella et al., 2018*).

Unsigned networks are widely applied in fields such as social sciences, biology, and transportation, where connections are essential but do not inherently carry positive or negative attributes. Research in unsigned networks often focuses on understanding the overall structure of the network, detecting communities, predicting links, and analyzing the roles of specific links and nodes, such as bridges. These studies uncover patterns, optimize performance, and reveal hidden structures in complex systems.

## Signed networks

Understanding the distinctions between signed and unsigned networks is essential for bridge analysis. Unlike unsigned networks, which indicate connections without additional context, signed networks explicitly represent positive or negative interactions between nodes. Positive interactions signify cooperative or friendly relationships, while negative interactions indicate conflict or antagonism. This distinction provides deeper insight into the structural and functional roles of bridges in networks (*Brzozowski, Hogg & Szabo, 2008*; *Kunegis, Lommatzsch & Bauckhage, 2009*; *Lampe, Johnston & Resnick, 2007*).

Formally, a signed network is defined as $G = (V, E^+, E^-)$, where $V$ denotes the set of nodes, $E^+$ represents the set of positive edges indicating friendly relationships, and $E^-$ represents the set of negative edges indicating antagonistic relationships. Positive edges in $E^+$ are assigned values greater than 0. Negative edges in $E^-$ are assigned values less than 0.

Modeling both positive and negative interactions offers a comprehensive framework for understanding real-world social networks. Signed networks capture broader social dynamics than unsigned networks by incorporating both cooperative and adversarial relationships. Positive ties reflect trust or alliances, whereas negative ties highlight conflict or rivalry. This dual nature of relationships introduces complexities for algorithms originally designed for unsigned networks, which often struggle with negative links (*Girdhar & Bharadwaj, 2017*; *Kim et al., 2023*). Consequently, specialized approaches are required for tasks such as community detection, link prediction, and influence modeling in signed networks to effectively account for both cooperative and adversarial interactions (*Yang et al., 2012*; *Wang et al., 2017*).

## Global and local bridges

This section categorizes bridges based on the structural impact of each type: (i) global bridges and (ii) local bridges. A global bridge is an edge whose removal results in the disconnection of the networks into separate components (*Huang et al., 2019a*). As shown

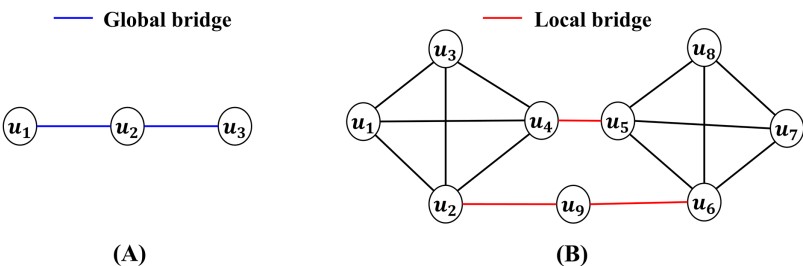

**(A)**                                      **(B)**

**Figure 1** **(A) Global bridge (blue): Removing bridge disconnects the network. (B) Local bridge (red): Removing bridge increases the shortest path but keeps the network connected.**

in Fig. 1A, the blue edge between $u_1$ and $u_2$ represents a global bridge. Its removal divides the network into two disconnected components. *Jensen et al. (2016)* used global bridges to identify critical nodes, such as airports, within air transportation networks. Additionally, global bridges have been used to identify interdisciplinary nodes in scientometric networks, highlighting their role in facilitating connectivity across diverse domains.

In contrast, a local bridge is an edge whose removal increases the shortest path distance between its endpoints to a value greater than 2, without disconnecting the network (*Granovetter, 1973*). For example, in Fig. 1B, the red edges $\{u_4, u_5\}$ and $\{u_2, u_9\}$ are local bridges. Removing these edges increases the distance between their endpoints, but does not split the network into separate components. *Papadopoulos et al. (2009)* demonstrated the utility of local bridges in community detection, where they are used to identify boundaries between groups within networks. Methods utilizing local bridging values have been applied to delineate community structures in real-world networks, such as user-generated platforms. For simplicity, this article collectively refers to global and local bridges as bridges.

## Importance of bridges

Bridges (*Bondy & Murty, 2008*; *Burt, Kilduff & Tasselli, 2013*; *Burt & Soda, 2021*) are critical for understanding the structure and functionality of the network, particularly for analyzing the robustness and resilience of networks. Bridge nodes (or articulation points) are nodes whose removal increases the number of connected components, often linking different subgroups and ensuring network cohesion.

Bridge nodes play a pivotal role in extending the reach of 2-hop connections, particularly when compared to non-bridge nodes. As shown in Fig. 2A, information originating from a non-bridge node remains confined to a single community. In contrast, Fig. 2B illustrates how a bridge node facilitates connections between multiple communities, with blue lines representing bridges. This illustrates the importance of bridge nodes in expanding connectivity and linking disparate parts of the network. *Zhang & Li (2017)* further underscore the importance of bridge nodes in enabling the diffusion of information and rumors between communities, highlighting their role as critical pathways in social networks.

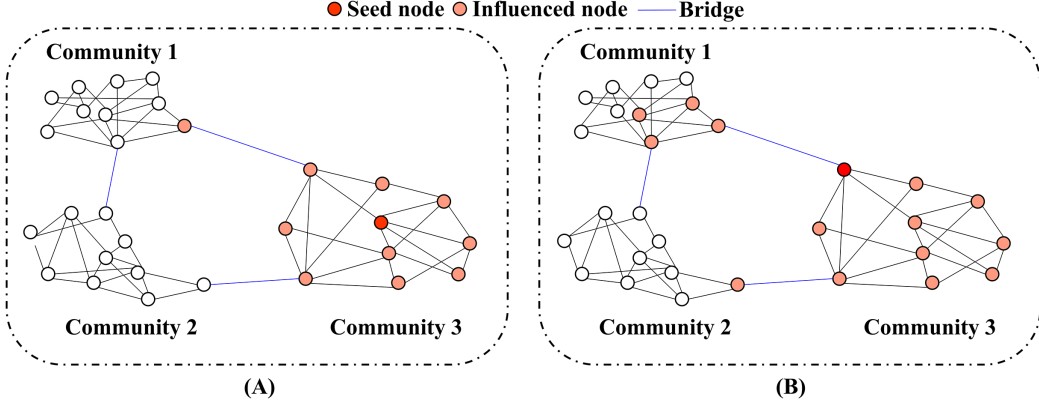

**Figure 2** (A) 2-hop neighborhood from a non-bridge node and (B) from a bridge node, where the seed node (red) influences nearby nodes (orange) within 2 hops. Blue edges represent bridges connecting communities.

**Table 2** Research areas categorized by bridge types in unsigned networks.

| Article | Year | Structural | Functional |
|---|---|---|---|
| *Granovetter (1973)* | 1973 | ✓ | |
| *Burt (2005)* | 1992 | | ✓ |
| *Hwang et al. (2008)* | 2008 | ✓ | |
| *Papadopoulos et al. (2009)* | 2009 | ✓ | |
| *Musiał & Juszczyszyn (2009)* | 2009 | ✓ | |
| *Salathé & Jones (2010)* | 2010 | | ✓ |
| *Pescher & Spann (2014)* | 2014 | | ✓ |
| *Gao, Musial & Gabrys (2017)* | 2017 | ✓ | |
| *Gryllos, Makris & Vikatos (2017)* | 2017 | | ✓ |
| *Zhang & Li (2017)* | 2017 | | ✓ |
| *Chen et al. (2018)* | 2018 | ✓ | |
| *Hu & Dong (2019)* | 2019 | ✓ | |
| *Corradini et al. (2020)* | 2020 | ✓ | |
| *Vikatos, Gryllos & Makris (2020)* | 2020 | | ✓ |
| *Chen et al. (2020)* | 2020 | | ✓ |
| *Meghanathan (2021)* | 2021 | ✓ | |
| *Lin et al. (2021)* | 2021 | | ✓ |
| *Burt & Soda (2021)* | 2021 | | ✓ |
| *Wang et al. (2022)* | 2022 | | ✓ |
| *Gao et al. (2023)* | 2023 | | ✓ |

# BRIDGE PROPERTIES

This section examines different types of bridge and their specific applications. Tables 2 and 3 present studies on bridges in unsigned and signed networks, organized by publication year. Each study is classified according to its focus on structural or functional bridge types.

**Table 3 Research areas categorized by bridge types in signed networks.**

| Article | Year | Structural | Functional |
|---|---|---|---|
| *Zhang et al. (2016)* | 2016 | | ✓ |
| *Ehsani & Mansouri (2023)* | 2019 | ✓ | |
| *Beers et al. (2023)* | 2023 | | ✓ |
| *Wu et al. (2023)* | 2023 | ✓ | |

Structural bridges ensure network connectivity, while functional bridges support efficient information flow. Some bridges operate in both roles depending on the context.

**(i) Structural Bridges**: structural bridges also play a pivotal role in tasks such as graph representation and graph partitioning. In graph representation, these bridges help ensure that essential inter-cluster connections are preserved when high-dimensional graph data are embedded into lower dimensions, which is critical for tasks like node classification and link prediction. Methods such as DeepWalk (*Perozzi, Al-Rfou & Skiena, 2014*) and node2vec (*Grover & Leskovec, 2016*) explicitly emphasize the preservation of structural features, including bridges, during the embedding process. Bridge edges tend to increase the probability that random walks traverse between communities, thereby allowing embedding algorithms such as DeepWalk and node2vec to better capture inter-community relationships that might otherwise be underrepresented in proximity-based sampling. In graph partitioning, structural bridges minimize inter-cluster disconnections, maintaining the coherence of community structures. Approaches such as the Louvain method (*Blondel et al., 2008*) and modularity optimization techniques (*Newman, 2006*) rely on these bridges to identify well-defined clusters, thus ensuring the integrity of the overall network topology.

**(ii) Functional Bridges**: functional bridges enable communication and resource flow between network regions, fostering cross-community interaction and collaboration. Unlike structural bridges, which primarily maintain network connectivity, functional bridges optimize the distribution of information and resources, enabling efficient exchanges between otherwise isolated regions (*Burt, 2005*). These bridges are essential for facilitating information propagation across networks. Functional bridges enable efficient transfer of knowledge, influence, and resources between communities, often driving the success of processes such as viral marketing and rumor diffusion. Studies on influence maximization (*Kempe, Kleinberg & Tardos, 2003*) have demonstrated how functional bridges accelerate information dissemination by linking disconnected network components. Similarly, epidemic models (*Pastor-Satorras & Vespignani, 2001*) highlight the role of these bridges in maintaining connectivity during the spread of critical information. By optimizing communication dynamics, functional bridges enhance network-wide interaction and ensure effective propagation in dynamic and large-scale systems.

## RESEARCH AREAS

This section introduces three main research areas relevant to the study of bridges in social networks. These areas are categorized into graph representation and graph partitioning,

which are associated with structural bridges, and information propagation, which pertains to functional bridges. Key concepts, methodologies and significant findings are discussed, with a distinction made between studies on signed and unsigned social networks.

## Structural bridges

Graph representation and partitioning focus on the role of structural bridges in preserving network connectivity and identifying community structures. Structural bridges are crucial to maintaining the overall structure of the network by preserving connections between subcomponents.

### Graph representation

Graph representation, also called network embedding, maps nodes to a low-dimensional space while preserving the structural properties of the network. This representation facilitates various network analysis tasks, such as node classification, link prediction, and visualization. In addition, graph clustering algorithms use embeddings to reduce network complexity and enable efficient analysis (*Perozzi, Al-Rfou & Skiena, 2014*; *Tang et al., 2015*; *Grover & Leskovec, 2016*; *Wang et al., 2018*).

In unsigned networks, understanding the roles of bridges is crucial for accurate network embedding and clustering. *Musiał & Juszczyszyn (2009)* investigated the properties of bridges in social networks, focusing on how bridges connect different clusters within a network and affect embedding. Their analysis demonstrated that bridges are essential to maintain network connectivity and facilitate efficient information flow. The study also showed that removing bridges often leads to network fragmentation, underscoring their role in preserving network cohesion.

*Corradini et al. (2020)* defined and detected $k$-bridges in social networks, using the Yelp dataset as a case study. A $k$-bridge is an edge whose removal increases the number of connected components by $k$ or more. This concept extends the traditional notion of bridges by evaluating the impact on network connectivity with a finer level of granularity. The authors developed efficient algorithms to identify $k$-bridges, revealing the underlying structure and intrinsic community organization within social networks. The findings highlighted the importance of $k$-bridges for clustering nodes and understanding community structures, which are essential for robust network embedding tasks.

In signed networks, *Chen et al. (2018)* improved embedding in signed directed networks by identifying and leveraging bridges. The proposed bridge-enhanced embedding technique focused on detecting bridges to maintain network connectivity. Bridges were defined by their potential to fragment the network when removed, as their removal significantly increased the distance between other nodes. The detection process combined network topology analysis with heuristic algorithms. During the embedding process, greater importance was assigned to bridges and the edges directly connected to them.

*Wu et al. (2023)* developed the Signed Directed Attention Network (SDAN) model to address challenges related to bridges in signed directed networks. The SDAN model employs a multi-head attention mechanism to adjust node importance based on edge direction and sign. The model highlights the role of bridges in maintaining connectivity by

**Peer**J Computer Science

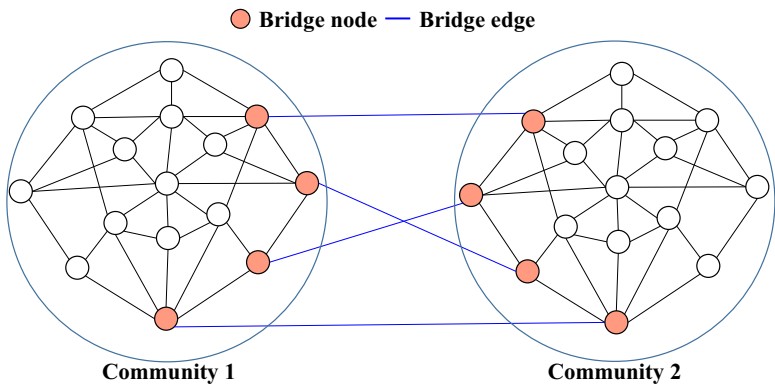

**Figure 3** Bridge nodes and bridge edges divide the graph into distinct communities, with bridges representing links between communities. 

assigning higher weights to them during the encoding process, capturing key structural features. In addition, the model integrates balance and status theories to better understand the dynamics of the network. The embeddings preserve essential network characteristics and model complex relationships within signed networks.

### Graph partitioning

Graph partitioning, or community detection, involves dividing a graph into smaller subgroups or communities. A common approach is edge-cutting, where edges are removed to reduce connectivity and separate the network. Edge selection is typically based on measures such as centrality or the role of edges in connecting communities. The removal of selected edges highlights the structure of the network by separating densely connected regions. Bridges, which connect distinct communities, are often targeted to identify community boundaries, which makes them essential for community detection. For example, Fig. 3 illustrates the graph partitioning process, with a focus on bridges between different communities. The bridges, highlighted in orange, play a key role in linking the communities, while the blue lines represent weaker connections facilitated by these bridges. The figure demonstrates how bridges contribute to maintaining connectivity between communities.

In unsigned networks, *Papadopoulos et al. (2009)* introduced the Bridge Bounding method, a localized approach for efficient community discovery. In this approach, bridges are defined as nodes that link densely connected subgroups. The Bridge Bounding algorithm identifies bridges by analyzing local connectivity and structure. It calculates edge betweenness centrality, which measures the number of shortest paths passing through a given edge. Nodes connected by edges with high betweenness are identified as potential bridges. These bridges define the boundaries of communities, reflecting the underlying topology of the network.

*Hu & Dong (2019)* proposed a cross-social-network interconnection model based on bridge communities. The model focuses on the role of bridges in connecting different social networks. It ranks nodes using centrality measures such as degree, betweenness, and closeness centrality. High-ranking nodes are matched across networks using similarity

measures based on user attributes, such as username, gender, and location. Nodes with high similarity form bridge communities that act as bridges between networks. The model strengthens the connections between these nodes to improve interconnectivity. This process facilitates the dissemination of information and maintains network connectivity. The framework employs clustering algorithms that account for the enhanced connectivity provided by bridges. The resulting clusters reflect the true interconnectivity of the networks. Although the primary objective of this approach is to enhance interconnectivity across social networks, the clustering component of the framework also contributes to graph partitioning by identifying community structures reinforced through bridge-based connections.

Hwang et al. (2008) introduced bridging centrality, a novel metric designed to identify nodes and edges that serve as critical bridges between network communities. Bridging centrality combines betweenness centrality with a bridging coefficient, which measures the likelihood of connecting distinct network modules. This combination captures both local and global properties, highlighting key elements that maintain network cohesion and facilitate information flow. The article also presents a graph clustering algorithm that leverages the bridge centrality to define community boundaries. By iteratively removing edges with the highest bridging centrality scores, the algorithm identifies clusters with higher modularity and effectively distinguishes cohesive substructures within the network.

Gao, Musial & Gabrys (2017) proposed a Community Bridge Boosting Prediction Model (CBBPM), which improves link prediction by using community bridges. The model categorizes the nodes as within-community or cross-community, depending on whether they connect different communities. The core idea is to boost the similarity scores of the bridges, defined as nodes connecting different communities. Bridges are identified based on their degree and the proportion of connections to different communities. Nodes with low degrees or those most closely connected to a single community are excluded. Boosting the similarity scores of the selected bridges improves their importance in the prediction model, resulting in improved link prediction accuracy across various social networks. Figure 4 illustrates the crucial role of bridges in the CBBPM model. The red nodes represent bridges that facilitate connections between communities, while the blue lines show the predicted potential links between these communities, demonstrating how bridges enhance link prediction.

In signed networks, Ehsani & Mansouri (2023) introduced the BridgeCut algorithm to partition signed networks into balanced subgraphs. The algorithm ensures a fair and efficient partitioning process. BridgeCut, inspired by the Girvan-Newman (Blondel et al., 2008) community detection method, focuses on negative edges that act as bridges. It calculates edge betweenness centrality to identify negative edges functioning as bridges. These edges are iteratively added to the bridge set to improve the network's balance index. The process divides the network into two near-balanced partitions, with the remaining nodes assigned based on their connections. BridgeCut was evaluated on multiple datasets, achieving balanced partitions with high internal stability and a low proportion of negative edges.

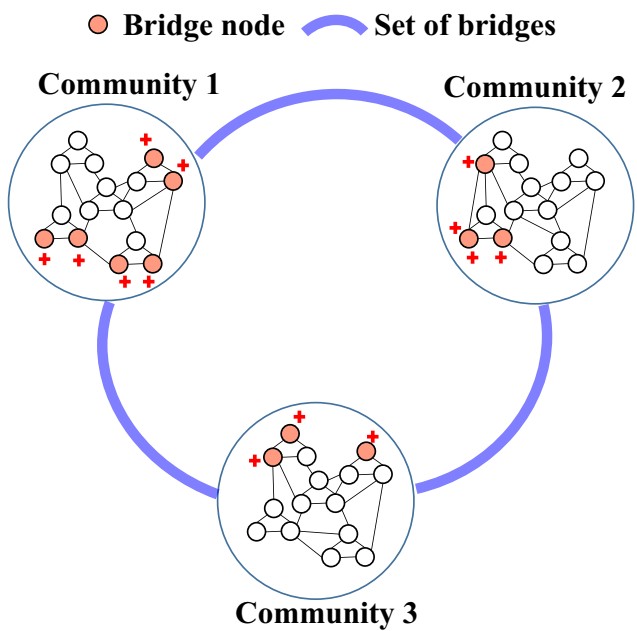

**Figure 4** Representation of bridges and their role in enhancing link prediction across communities.

## Functional bridges

Information propagation examines how functional bridges support the flow of information between communities. These bridges improve communication and resource distribution between disconnected regions, ensuring efficient interaction across communities. Table 4 summarizes key studies in these research areas, categorizing them by bridge types (structural or functional) and highlighting their main focus of research.

### *Information propagation*

*Salathé & Jones (2010)* analyze the dynamics of disease spread in networks with community structures. The study identifies community bridges as nodes that connect different communities and play a critical role in disease transmission. The authors show that targeting community bridges for immunization is more effective in reducing the spread of diseases than targeting nodes with high centrality. To identify these nodes efficiently, they propose a Community Bridge Finder (CBF) algorithm, which uses local network information.

*Zhang & Li (2017)* investigate the propagation of rumors in networks with community structures, focusing on the role of bridge hubs. The study identifies bridge hubs as nodes with high connectivity within and between communities, which significantly influence the speed and scale of rumor dissemination. The authors show that selecting bridge hubs as initial spreaders leads to faster and broader propagation compared to other nodes, highlighting their importance in cross-community information dissemination.

*Gao et al. (2023)* propose a multi-feature rumor suppression mechanism that incorporates community division. The study identifies bridges connecting different

**Table 4 Research areas categorized by bridge types.**

| Bridge type | Article | Research focus |
|---|---|---|
| Structural | *Granovetter (1973)* | Role of weak ties in connecting isolated communities |
| | *Hwang et al. (2008)* | Bridging centrality to detect key community links |
| | *Papadopoulos et al. (2009)* | Identifying boundaries between communities *via* bridges |
| | *Musiał & Juszczyszyn (2009)* | Analyzing bridges in network partitioning |
| | *Gao, Musial & Gabrys (2017)* | Understanding community structure through bridge links |
| | *Chen et al. (2018)* | Embedding methods emphasizing inter-community links |
| | *Hu & Dong (2019)* | Cross-network community interconnections *via* bridges |
| | *Corradini et al. (2020)* | K-bridge detection for clustering and community boundaries |
| | *Meghanathan (2021)* | Neighborhood-based centrality for identifying bridges |
| | *Ehsani & Mansouri (2023)* | Balanced network partitioning using BridgeCut |
| | *Wu et al. (2023)* | Embedding-based signed bridge detection |
| Functional | *Burt (2005)* | Information brokerage through structural holes |
| | *Salathé & Jones (2010)* | Spread of diseases through community bridges |
| | *Pescher & Spann (2014)* | Product-related information diffusion across groups |
| | *Gryllos, Makris & Vikatos (2017)* | Bridges for optimizing marketing reach |
| | *Zhang & Li (2017)* | Rumor propagation through cross-community bridges |
| | *Vikatos, Gryllos & Makris (2020)* | Cross-community message dissemination in multiplex networks |
| | *Chen et al. (2020)* | Efficient information propagation using iBridge links |
| | *Lin et al. (2021)* | Role of structural holes in information dissemination |
| | *Burt & Soda (2021)* | Organizational information exchange through brokers |
| | *Wang et al. (2022)* | Social bridges sharing brand content |
| | *Gao et al. (2023)* | Rumor suppression using multi-feature bridges |
| | *Zhang et al. (2016)* | Trust hole identification for conflict resolution |
| | *Beers et al. (2023)* | Bridges connecting polarized communities during elections |

communities as key targets for rumor suppression. The proposed algorithm selects these bridges to effectively block rumors, reducing their influence across the network. The simulation results demonstrate that prioritizing these bridges improves the suppression efficiency.

*Chen et al. (2020)* proposed the iBridge model, which improves link prediction by using community detection and positive unlabeled (PU) learning to detect bridge links that connect nodes across communities, allowing information propagation between disconnected groups. Using PU learning, the model effectively handles a predominance of positive examples and numerous unlabeled cases, improving the accuracy of bridge link predictions. The integration of community-specific features and a robust framework allows iBridge to significantly refine inter-community link prediction, outperforming traditional methods that neglect community structure. Figure 4 demonstrates the focus of the iBridge model on bridge links (shown as blue lines) that connect nodes in different communities (orange nodes), thus facilitating information diffusion and improving link prediction accuracy.

*Pescher & Spann (2014)* examine the role of bridging positions in product-related information diffusion. The study reveals that actors in bridging positions access diverse information sources and influence cross-cluster knowledge transfer. However, these actors are less likely to share product-related information, suggesting that bridging positions may not significantly contribute to the spread of marketing messages.

*Gryllos, Makris & Vikatos (2017)* present a methodology to extract bridges from social networks to optimize marketing campaign targeting. Bridges, defined as nodes that link disparate communities within the network, are essential to improve information flow and maximize the reach of marketing messages. The authors propose a process that involves identifying influential nodes, detecting bridges, and leveraging these bridges to amplify the spread of marketing campaigns. This approach emphasizes the importance of bridges in maintaining network connectivity and facilitating the dissemination of information across different parts of the network.

*Vikatos, Gryllos & Makris (2020)* propose a methodology to improve the targeting of the marketing campaign by identifying bridges within multiplex social networks. Bridges are critical for linking different communities and facilitating the flow of information across diverse groups. The methodology involves several key steps: social network crawling to collect data, community detection and ranking to identify influential communities, link prediction to anticipate potential connections, and campaign targeting to optimize message dissemination. By analyzing bridges across multiple network layers, the authors demonstrate the pivotal role these nodes play in efficiently spreading marketing messages.

*Wang et al. (2022)* examine the role of social bridges in content sharing between distinct groups within social networks. Their study demonstrates that individuals with high betweenness centrality are more likely to share brand content, influenced by motivations such as self-identity signaling. In contrast, these individuals are less likely to share deal content due to its lower exclusivity and relevance to self-presentation. The findings suggest that the position of social bridges within a network affects the type of content they are motivated to share, highlighting the nuanced role of the network structure in content dissemination.

*Lin et al. (2021)* provides a comprehensive overview of the structural hole (SH) theory, which posits that individuals who bridge structural holes between different communities gain significant advantages by controlling the flow of information. These individuals, termed SH spanners, play a crucial functional role in social networks by acting as brokers that connect otherwise disconnected groups. The review highlights how SH spanners benefit from access to diverse information and opportunities, thus enhancing their social capital and influence within the network. The article categorizes SH spanner detection algorithms into two main types: information flow-based algorithms and network centrality-based algorithms.

*Burt & Soda (2021)* expand on this concept by examining the role of brokerage within social networks, particularly in bridging structural holes. Brokers serve as intermediaries, facilitating the exchange of information and resources between otherwise disconnected groups. Similarly to SH spanners, brokers fulfill a functional role by enabling interaction and collaboration among separated communities within a network. In signed networks,

bridges, like brokers, are responsible not only for managing positive exchanges but also for mitigating conflicts, ensuring the network's operational stability and adaptability. This makes bridges, or functional bridges, crucial for sustaining the coherence of the network and fostering innovation between diverse groups.

Further supporting these ideas, *Beers et al. (2023)* analyze co-engagement networks during the 2020 US election, emphasizing the role of bridges in maintaining interactions between polarized communities. These bridges facilitate communication between politically distinct groups, such as supporters of opposing parties, by enabling the flow of information and perspectives. In signed networks, bridges similarly bridge divided communities, helping to preserve network cohesion while managing potential conflicts. Their function is critical in maintaining functional bridges that support both information exchange and conflict resolution between various groups.

The article by *Zhang et al. (2016)* on trust holes in signed networks explores the role of specific nodes in bridging trust and distrust between communities. Trust holes are identified as critical structures that mediate interactions between polarized groups by connecting otherwise disconnected regions within the network. Using the SCROLL framework, the study quantitatively evaluates the impact of trust holes on community cohesion and interaction costs, providing a robust mechanism to optimize community detection and reduce conflict. Furthermore, the findings demonstrate that removing trust holes can significantly fragment the network, leading to weakened inter-community connectivity and disrupted information flow.

# APPLICATIONS IN OTHER DOMAINS

The concept of bridges, as discussed in earlier sections, is studied in various fields. This section examines their roles in medical networks, health and behavioral sciences, and biological networks.

## Medical networks

Bridge symptoms function as critical bridges in psychological networks, facilitating the co-occurrence and interaction of mental disorders by linking distinct clusters of symptoms. These functional bridges maintain connectivity within disorder clusters, enabling interactions that perpetuate comorbid conditions. *Groen et al. (2020)* explored the comorbidity between depression and anxiety using a dynamic psychological network perspective. The study identified bridge mental states, such as "worrying" and "feeling irritated," which act as critical links connecting distinct symptom clusters. These bridge symptoms facilitate the persistence of comorbidity and exacerbate both conditions over time, highlighting their potential as targets for therapeutic interventions.

*Jones, Ma & McNally (2021)* introduced the concept of bridge centrality to quantify the importance of symptoms that connect different mental disorders within symptom networks. Four network metrics—bridge strength, bridge betweenness, bridge closeness, and bridge expected influence—were developed to measure these bridges. Applying these metrics to 18 group-level comorbidity networks demonstrated how bridge symptoms can drive the progression from one disorder to another. Together, these studies underscore the

value of identifying bridge symptoms to inform intervention strategies and improve mental health outcomes.

## Health and behavioral sciences

Recent studies have used network analysis to identify bridge symptoms and traits that connect diverse psychological and behavioral conditions. *Zhao et al. (2023)* modeled relationships between internet addiction (IA) and depression, identifying bridge symptoms such as "guilty" and "escape" that link symptom clusters across both conditions. *Qu et al. (2024)* applied cross-lagged panel network analysis to examine associations between short video addiction and depressive symptoms, highlighting "conflict" and "sad mood" as key bridges connecting these domains.

*Blasco-Belled (2023)* analyzed the interplay between character strengths and mental health indicators, identifying traits like "love", "hope", and "gratitude" as bridge nodes connecting positive and negative aspects of mental health. *Jurado-González et al. (2024)* investigated comorbidity among anxiety, depression, and somatic symptoms, showing how bridge symptoms such as "low energy" and "sad mood" sustain co-occurring conditions. These insights emphasize that targeting bridge symptoms and traits can disrupt maladaptive interactions and support more effective interventions.

## Biological networks

In unsigned networks, *Cheng, Huang & Sun (2008)* defined bridge motifs as weak links connecting distinct modules in biological systems. Their detection method uncovers global statistical properties and local connection structures that are functionally significant. Applied to gene regulation networks, bridge motifs were shown to highlight topological patterns distinguishing species.

*Panni & Rombo (2015)* proposed strategies for identifying disease modules composed of interconnected genes or proteins linked to specific diseases. These disease modules act as functional bridges between different disease pathways, aiding the understanding of comorbidities and revealing therapeutic targets. For example, bridges linking drug-target modules to disease clusters can inform repositioning efforts by identifying shared molecular mechanisms underlying multiple conditions. Together, these studies illustrate how bridge structures can uncover complex relationships among biological entities.

## Transportation networks

In transportation networks, bridges play a critical role in maintaining connectivity and mitigating cascading failures. *Xiao, Huang & Tang (2023)* proposed a quantitative framework to assess the importance and correlation of urban bridges and roads for evaluating road network vulnerability. The study introduced measurement indicators such as node degree, proximity centrality, intermediary centrality, and interrupt centrality to capture the structural significance of bridges. By modeling road networks using dual representations and applying grey correlation analysis, the authors demonstrated that bridge nodes often exhibit the highest correlation with overall network performance, underscoring their pivotal role in sustaining system resilience.

*Qiang, Wancheng & Xinzhi (2021)* focused on seismic vulnerability and retrofit prioritization of bridge networks under budget constraints. They developed multiple bridge importance ranking methods combining topological and fragility considerations and formulated the selection of retrofit strategies as a 0–1 knapsack optimization problem. Simulation results revealed that retrofitting strategies prioritizing both network topology and structural fragility achieved superior improvements in post-seismic reliability compared to methods relying solely on individual criteria. These findings highlight that the strategic reinforcement of critical bridges is essential for enhancing the resilience and reliability of transportation networks in seismic regions.

### Energy networks and power systems

In energy networks, bridges play an essential role in improving stability and preventing cascading failures. *Lan & Zocca (2021)* proposed a bridge-block decomposition refinement framework that partitions power grids into smaller sub-networks by selectively switching off transmission lines. This process increases the number of bridge-blocks, which act as barriers to failure propagation, thereby improving robustness against cascading line failures. The authors introduced a novel Mixed-Integer Linear Programming (MILP)-based optimization approach using DC power flow models, as well as an efficient AC recursive tree partitioning method, demonstrating that these strategies can significantly enhance failure localization while minimizing line congestion.

*Ódor et al. (2024)* analyzed European high-voltage power grids using large-scale Kuramoto swing equation simulations to study how adding or removing bridges affects network synchronization and cascade dynamics. Their results show that well-placed bridges improve synchronization and reduce cascade sizes near the synchronization transition region, where self-organization drives power grid behavior. Conversely, bridge removals or poorly placed additions can exacerbate failures due to Braess's paradox. The study emphasizes that bridge placement must consider both topological and dynamical factors to achieve optimal resilience and prevent large-scale blackouts.

## OPPORTUNITIES AND CHALLENGES

In this section, we explore the opportunities and challenges presented by signed and unsigned networks, which are crucial for advancing the field of network analysis.

### Unsigned networks

In unsigned networks, which lack edge signs but may still include edge weights, local bridges are defined as edges whose removal increases the shortest path between two nodes by at least two (*Granovetter, 1973*). However, this definition implicitly assumes unweighted graphs, treating all connections as equally important regardless of their strength. In practice, bridges causing larger increases in effective distance or occupying more central structural positions can contribute more significantly to network connectivity and information flow. This oversimplification reduces the ability to effectively evaluate the diverse roles of bridges that strengthen community cohesion and facilitate information propagation.

This issue is not limited to local bridges but applies to broader bridge definitions. Equal treatment of all bridges overlooks differences in their contributions to network functionality, which depend on factors such as network position and the relative significance of the connections. Incorporating metrics that reflect these varying contributions could improve the assessment of community cohesion, information dissemination, and network resilience (*Opsahl, Agneessens & Skvoretz, 2010*).

Beyond the foundational definition by *Granovetter (1973)*, recent studies have extended the analysis of network connectivity and robustness by incorporating weighted metrics and meso-scale structures. *Opsahl, Agneessens & Skvoretz (2010)* introduced measures accounting for both edge weights and connectivity patterns, while *Rombach et al. (2017)* proposed a framework for detecting core-periphery structures, which can reveal cohesive cores critical for maintaining overall connectivity. However, despite these advances, there remains a need to integrate bridge-centric and core-periphery perspectives to develop unified metrics that can better quantify the role of critical connections in supporting resilience, information diffusion, and structural cohesion in complex networks.

Building upon applications described in prior sections (*Xiao, Huang & Tang, 2023*; *Qiang, Wancheng & Xinzhi, 2021*), recent studies have demonstrated the importance of identifying critical bridges to improve resilience in transportation networks. However, most existing approaches remain static and do not consider temporal fluctuations in traffic loads or network topology. Future research could develop predictive models that integrate structural indicators with real-time flow data to dynamically assess bridge vulnerability and support proactive intervention strategies under disruptions.

## Signed networks

Signed networks reflect dual relationships with positive and negative edges (*Tang et al., 2016*; *Leskovec, Huttenlocher & Kleinberg, 2010*; *Yang et al., 2012*). Structural bridges in signed networks influence inter-community dynamics by facilitating cooperation through positive connections or highlighting conflicts through negative connections. These characteristics distinguish signed networks from unsigned networks and highlight the need to reconsider how key structural elements are defined (*Yang, Cheung & Liu, 2007*; *Derr, Ma & Tang, 2018*).

The definition of bridges in unsigned networks, which focuses solely on maintaining connectivity, cannot be directly applied to signed networks due to the duality of positive and negative edges. Algorithms designed for unsigned networks do not capture the unique complexities of signed relationships (*Kunegis et al., 2010*). The duality of edges complicates the identification of structural elements, such as bridges. Table 3 illustrates the absence of a widely accepted definition of bridges in signed networks. Balance Theory (*Heider, 1946*) and Status Theory (*Leskovec, Huttenlocher & Kleinberg, 2010*) offer foundations for analyzing signed relationships but lack integration into bridge definitions.

Recent advances have proposed signed graph neural networks and embedding methods (*Derr, Ma & Tang, 2018*; *Huang et al., 2019b*; *Liu, 2022*; *Zhou & Yan, 2025*), which aim to capture the dual nature of positive and negative relationships. However, these approaches primarily focus on node classification and link prediction, with limited attention to the

identification and characterization of structural bridges. Developing a comprehensive framework that integrates sign-aware embedding with structural metrics could improve the quantification of bridge importance and enable tracking of their temporal evolution. This refined perspective may support practical applications such as polarization mitigation, conflict prediction, and resilience assessment in online social platforms.

### General considerations for bridge analysis

In addition to the specific challenges in unsigned and signed networks, several general considerations apply to both types of networks. Bridges in directed networks require an analysis of edge directionality, as their removal may impact flows in one direction while preserving the opposite flow (*Malliaros & Vazirgiannis, 2013*). In weighted networks, the importance of bridges depends on edge weights, where high-weight bridges often play more significant roles in information diffusion (*Opsahl, Agneessens & Skvoretz, 2010*; *Riascos & Mateos, 2021*). Bridge detection in large-scale networks requires algorithms that remain computationally efficient while processing large datasets (*Perozzi, Al-Rfou & Skiena, 2014*; *Harenberg et al., 2014*; *Tang et al., 2015*). Furthermore, time-varying networks introduce additional complexity, which requires adaptive algorithms that can efficiently handle real-time updates at nodes and edges (*Xue et al., 2022*; *Rossetti & Cazabet, 2018*).

Recent advances in dynamic graph learning (*Pareja et al., 2020*; *Chen et al., 2024*) have introduced frameworks capable of modeling evolving topologies and time-dependent interactions, which could be adapted to improve bridge detection in temporal networks. Future research may explore the integration of dynamic embedding techniques with bridge importance metrics to support applications such as early warning systems in infrastructure networks and adaptive content moderation in online platforms.

## CONCLUSION

In this article, the concept of bridges in social networks was systematically reviewed. The discussion highlighted both structural and functional roles of bridges. Findings across diverse domains were synthesized to provide a comprehensive perspective. Consolidation of existing research clarifies conceptual boundaries and reveals open questions for future work. For example, bridge-aware embedding methods represent promising directions for improving community detection and modeling information diffusion. Dynamic bridge analysis may also support the understanding of evolving network resilience. This survey aims to provide a foundation for developing advanced models and tools to analyze bridges and their impact on social and information networks.

### Funding

This research was supported by the research fund of Chungnam National University. The APC was funded by Institute of Information & communications Technology Planning & Evaluation(IITP) grant funded by the Korea government (MSIT) (No. RS-2022-00155857,

Artificial Intelligence Convergence Innovation Human Resources Development (Chungnam National University)) and by the BK21 FOUR Program by Chungnam National University Research Grant, in 2024. The funders had no role in study design, data collection and analysis, decision to publish, or preparation of the manuscript.

## Grant Disclosures

The following grant information was disclosed by the authors:
Chungnam National University.
Institute of Information & Communications Technology Planning & Evaluation (IITP)
Korea Government (MSIT): RS-2022-00155857.
Artificial Intelligence Convergence Innovation Human Resources Development.
BK21 FOUR Program by Chungnam National University Research Grant.

## Competing Interests

The authors declare that they have no competing interests.

## Author Contributions

- Jeongseon Kim conceived and designed the experiments, performed the experiments, analyzed the data, prepared figures and/or tables, authored or reviewed drafts of the article, and approved the final draft.
- Soohwan Jeong analyzed the data, prepared figures and/or tables, authored or reviewed drafts of the article, and approved the final draft.
- Jungeun Kim analyzed the data, authored or reviewed drafts of the article, and approved the final draft.
- Sungsu Lim analyzed the data, authored or reviewed drafts of the article, and approved the final draft.

## Data Availability

This is a literature review.

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
