# Peer review of "Bridges in social networks: current status and challenges"

_PeerJ Computer Science, doi:10.7717/peerj-cs.3122_

## Round 0.1 · original submission · Major Revisions

Dear authors

Thank you for submitting your manuscript entitled "Bridges in Social Networks: Current Status and Challenges" to PeerJ Computer Science. The topic addressed in your review—bridge detection and analysis in social networks—is timely and relevant to a broad audience within the journal’s scope.

After careful consideration of the manuscript and the detailed comments provided by the reviewers, I have decided that the paper has potential, but also requires substantial revisions before it can be reconsidered for publication. Therefore, I am issuing a decision of Major Revision.

The reviewers agree that the manuscript covers a relevant and underexplored topic, but they also highlight important aspects that should be addressed to improve its clarity, coherence, and overall impact. Among the most pressing concerns is the need to clarify the conceptual scope, particularly the distinction between social networks and other types of networks, as well as the definition and role of functional bridges in contrast to functional networks. The survey methodology also requires more transparency, including clearer criteria for the selection and filtering of literature, as well as consideration of the scientific impact of the referenced works. Reviewers point out inconsistencies in citation formatting and a lack of justification for including certain older sources while omitting more recent and influential ones. They also suggest a more refined and logical organization of sections, especially in relation to bridge properties, their role in graph partitioning and embedding models, and their connection to concepts such as structural holes, hubs, and communities. Furthermore, the discussion of challenges and future directions should be strengthened with up-to-date literature and more concrete examples, especially in applied domains like medical or biological networks. Finally, please ensure thorough proofreading to correct formatting issues, typographical errors, and improve overall presentation.

We invite you to submit a thoroughly revised version of your manuscript that carefully addresses the reviewers’ concerns. Please include a point-by-point response to each comment, indicating how you have revised the manuscript in response.

Your revised submission will undergo another round of review, possibly including the original reviewers.

We appreciate the effort you have put into this work and look forward to seeing an improved version of your manuscript.

**Language Note:** The review process has identified that the English language must be improved. PeerJ can provide language editing services - please contact us at [email protected] for pricing (be sure to provide your manuscript number and title). Alternatively, you should make your own arrangements to improve the language quality and provide details in your response letter. – PeerJ Staff

Reviewer 1 ·

Basic reporting

The manuscript presents a review on the role of bridges in social network analysis, categorizing them into structural and functional types, as well as an emphasis on the bridge property, related research on graphs, and broader applications in various domains.

1.1: The review is of broad and cross-disciplinary interest and within the scope of the journal.

1.2: There has been related reviews on network structure, community detection, structural hole theory, and graph representation learning in recent years. While some aspects of bridge structures have been discussed, this manuscript could still offer a novel and useful perspective on the integration of bridge structure with relevant research on graphs and a broader applications.

1.3: The Introduction is generally good.

Experimental design

2.1: The Survey Methodology can be improved with better transparency in terms of scope, filtering criteria, and inclusion logic.
- In section 2, what is the final number of articles included in the review? How did the authors further narrow the scope of literature and refine the cognitive focus of the survey to ensure the "unbiased coverage"?

2.2: The manuscript lacks parentheses of citations in the main text.

2.3: Overall, the review is organized logically into coherent paragraphs. However, there are several questions here:
- In Section 3, are the authors assuming the audience has basic knowledge of nodes and edges but limited familiarity with directed, weighted, or signed networks? The logic in this section remains unclear. Could the authors clarify the structure and intent of introducing basic concepts for audience with broader accessibility?
- What is the relationship between bridges and other structural concepts in graphs, such as communities, hubs, and structural holes? A clearer comparison or framework in differentiating them would be helpful.

Validity of the findings

3.1: The argument is generally well developed and supported. However, there is still room for improvement.
- In Section 4, the transition from bridge properties to their role in graph partitioning and representation learning, such as embedding models of DeepWalk and node2vec, is not clearly articulated. Could the authors elaborate more on how structural bridge concepts relate to or play the role in these models?
- Why is the Hu and Dong's paper categorized under graph partitioning while it emphasizes improving interconnectivity in social networks?
- Some models have illustrative figures (e.g., Figure 4), while others do not. Is there a specific reason for emphasizing these works? If so, it would be helpful to explain the selection criteria for figure visualization.
- The literature review in Section 5.2 appears disorganized. For example, papers on betweenness centrality (Wang et al., 2022) and review on structural hole theory (Lin et al., 2021) are grouped under "Information Propagation" subsection. A more logical structure and refined categorization of works are more preferred.
- In Section 6, could the authors provide more concrete examples of how bridge structures function in medical and biological research? For instance, how might bridges help identify similarities or differences between drugs, symptoms, or diseases?
- In Section 7.1, do the authors mean "unweighted" networks rather than "unsigned" networks, when talking about "regardless of the magnitude" of importance of edge weights? Clarification is needed on referring to edge weights or edge signs.

3.2: The conclusion is good at identifying future directions, but could be made more specifically with concrete examples to fill the gaps.

Cite this review as

Reviewer 2 ·

Basic reporting

- The manuscript is written in professional, clear English and is accessible to a broad interdisciplinary audience.
- The introduction provides sufficient context and highlights the need for a systematic classification of bridges in social networks.
- The structure of the paper conforms to PeerJ’s standards and is logically organized, with clearly labeled sections and relevant figures/tables.
- Literature is well-cited, spanning both classical foundational works (e.g., Granovetter, Burt) and recent developments.
- One suggestion is to emphasize more clearly how this review differs from or improves upon previous surveys in the introduction, by explicitly stating the novelty and unique contributions of this work.
- Minor typos and formatting issues were found and should be corrected:
a) Line 28 (citation): “Freeman (7879)” → “Freeman (1978/79)” or “Freeman (1979)” (See the standard format of this journal).
Please ensure this correction is applied consistently across all occurrences of this citation in the manuscript.
b) Line 52: “bridge analyzes” → “bridge analyses”
c) Line 142: “wihtin” → “within”
d) Line 223: “Functional briges” → “Functional bridges”
- In addition, please carefully proofread the entire manuscript to identify and correct any remaining typographical or formatting errors that may have been overlooked.

Experimental design

- The authors applied a systematic and transparent literature search strategy, with inclusion and exclusion criteria clearly described.
- The survey scope is broad, covering both signed and unsigned networks and differentiating structural and functional bridge roles.
- The classification framework is well thought out and consistently applied.
- It may improve the clarity and novelty of the survey to include a short comparative summary of prior bridge-related reviews, if any exist.

Validity of the findings

- The framework presented (structural vs. functional bridges) is well-supported and appropriately referenced with examples across domains.
- Conclusions are logically derived from the discussion and reflect the structure laid out in the introduction.
- The paper also does a good job of identifying challenges and gaps in current research, particularly regarding signed networks.
- Adding a few lines about the potential real-world impacts of the findings would strengthen the final section.

Additional comments

- This is a comprehensive and well-organized literature review that will be useful to both new and experienced researchers in network science.
- The paper is clear and professional overall. Aside from the minor corrections noted above, no major issues were found.
- I recommend that the authors provide concrete examples to illustrate how this review can support real-world research, and explicitly demonstrate its practical impact, for instance in areas like community detection, information diffusion, infrastructure robustness (resilience), etc.

Cite this review as

Reviewer 3 ·

Basic reporting

The authors aim to present a review on an interesting and timely topic: the detection and analysis of bridge nodes in social networks, with a categorization into structural and functional bridges. I believe that a review on this subject is of broad interest and falls within the scope of the journal. To the best of my knowledge, no recent comprehensive review has been published on this specific topic.

However, in its current form, the manuscript offers a bibliographic compilation of relevant works without sufficiently justifying the impact of the topic or clearly outlining the challenges that remain for advancing scientific understanding in this area.

In the introduction, the authors present the topic and clearly state their motivation. Nevertheless, I have several comments regarding specific statements in the manuscript:

Lines 31–42: The manuscript describes various domains in which social networks are analyzed, citing biological and transportation networks. Given that social networks are typically defined as structures composed of individuals, groups, or organizations (nodes) connected by social relationships (edges), I believe referencing other types of networks in this context may lead to confusion. Although Section 6 addresses bridges in other domains, this paragraph blurs the focus by mixing social networks (which, as per the title, should be the central object of study) with other kinds of networks. While these may share abstract structural properties, they are conceptually distinct and should be treated accordingly.

Lines 63–71: The manuscript defines structural and functional bridges. I suggest that it would be useful to clarify—either here or in Section 3.1—that there is also a distinct concept of functional networks (e.g., correlation-based networks), which should not be confused with functional bridges, whose definition is tied to their role in information diffusion.

Lines 148–149: The statement “The focus in unsigned networks is primarily on the structural properties rather than the polarity of relationships” is, in my view, inaccurate. There is a substantial body of research—developed over the last decade by multiple research groups—that studies polarization in online social networks using unsigned weighted graphs. For example, see Morales et al., Chaos 25, 033114 (2015), DOI: [10.1063/1.4913758], which has received over 300 citations on Google Scholar.

Lines 180–181: The manuscript includes a claim regarding scientific collaboration networks. This statement should be supported by references to recent studies that validate it.

Experimental design

The review is organized in a coherent and logical manner. In the Survey Methodology section, the authors describe the approach used to select the articles included in this review, focusing primarily on keyword-based searches across various academic repositories.

However, this is where one of my strongest criticisms of the manuscript lies: the authors make no mention of the impact of the selected works. I believe that article impact should be a key criterion in any review process. This could include the total number of citations, citations per year, or other recognized metrics for evaluating scientific impact (e.g., Altmetric scores). It would also be helpful to know whether the impact factor of the journals was considered during the selection process.

In addition, the inclusion criterion based on publication year is unclear. The authors state that only works published from the year 2000 onward were considered, yet Tables 2 and 3 include earlier publications. This inconsistency should be clarified.

When citing references in the main text, the authors should carefully review Tables 2 and 3, as several references appear multiple times, and some are incorrectly formatted (e.g., “BurtBurt (2005)”). The citation format throughout the text should be improved for better readability—for instance, using square brackets [ ] would enhance clarity.

Finally, regarding the format of the references themselves, it is important to define acronyms and abbreviations used in conference or journal names. Readers may not be familiar with terms such as CIKM (Conference on Information and Knowledge Management), among others. Furthermore, it would be very helpful to include DOIs for the referenced works, enabling quick and direct access for the reader.

Validity of the findings

It is expected that, following a critical reading of the selected articles, the authors will be able to assess the current state of the topic and the challenges facing researchers. In addition to my critique of the selection methodology, I would like to make the following comments:

The Section 7: Opportunities and Challenges provides an opportunity for the authors to present, after critically reviewing the included works, their own contributions to advancing knowledge in the field. However, I believe this section could be improved, as the authors only offer very general considerations, often based on older works that are not necessarily recent. For example, in relation to unsigned networks (the most widely used), the citation used by the authors to propose improvements in the study of bridges is from 2010. A simple question on this topic to one of the most popular applications yields several highly relevant studies published in the last five years.. Similar comments can be made regarding the other subsections.

In the Conclusions section, it is stated that ongoing research and future advances in the field promise to unlock new potential and address existing challenges, paving the way for more robust and comprehensive social network models. While I agree with this statement, I do not believe that the manuscript provides sufficient explanation or evidence to adequately support this claim.

Additional comments

I believe the authors have identified a relevant and timely research topic: the fundamental role that bridges play in complex networks, particularly in social networks. The detection and analysis of “important” nodes within a network is a challenge that has been widely addressed in the literature for decades. In this regard, a bibliographic study that traces this trajectory and systematizes the accumulated knowledge to date would be valuable in guiding future research. However, in its current form, I do not believe this work fulfills that purpose for the reasons I have outlined in the previous sections.

Cite this review as

---

## Round 0.2 · accepted · Accept

The reviewers consider that the paper is now suitable for publication. Congratulations.

Reviewer 1 ·

Basic reporting

no comment

Experimental design

no comment

Validity of the findings

no comment

Additional comments

The authors addressed all my previous comments properly. I would recommend acceptance of this paper.

Cite this review as

Reviewer 2 ·

Basic reporting

The paper reads well and uses professional language throughout. I noticed the authors clarified the novelty in the introduction, which was helpful. Also, the earlier citation and formatting problems appear to have been addressed. Structurally, everything looks in order, including figures and tables. I think it meets the basic reporting criteria.

Experimental design

The paper follows a clear and systematic review process. The inclusion and exclusion criteria are described in reasonable detail, and the study selection appears unbiased. Citations are handled well, and the structure is easy to follow. I think the overall design is appropriate for a literature review.

Validity of the findings

The findings are consistent with the goals outlined in the introduction. The structural vs. functional bridge framework is well supported by the literature. The conclusion also outlines remaining challenges and future directions clearly.

Additional comments

The revised version is improved and It reads smoothly. I think the paper will be a helpful reference for those working on structural roles in social networks.

Cite this review as